# Oral Rabies Vaccine Strain SPBN GASGAS: Genetic Stability after Serial In Vitro and In Vivo Passaging

**DOI:** 10.3390/v14102136

**Published:** 2022-09-28

**Authors:** Stefan Borutzki, Benjamin Richter, Matthias Proemmel, Izabela Fabianska, Hon Quang Tran, Boris Hundt, Dietmar Mayer, Christian Kaiser, Andreas Neubert, Ad Vos

**Affiliations:** 1IDT Biologika GmbH, Am Pharmapark, 06861 Dessau-Rosslau, Germany; 2Ceva Innovation Center GmbH, Am Pharmapark, 06861 Dessau-Rosslau, Germany; 3TEW Servicegesellschaft mbH, Am Pharmapark 15, 06861 Dessau-Rosslau, Germany; 4Klocke Holding GmbH, Max-Becker-Str. 6, 76356 Weingarten, Germany

**Keywords:** NGS, back-passage, BHK21 Cl13, suckling mouse brain, reversion-to-virulence

## Abstract

Oral vaccination of wildlife has shown to be a very effective management tool in rabies control. Evaluation of the genetic stability of vaccine viruses before distributing vaccine baits in the environment is essential because all available oral rabies vaccines, including the genetically engineered rabies virus vaccine strain SPBN GASGAS (Rabitec), are based on replication-competent viruses. To evaluate the genetic stability of this vaccine strain, five serial passages of the Master Seed Virus (MSV) in the production cell line BHK21 Cl13 were performed. Furthermore, to test possible reversion to virulence, a back-passage study in suckling mouse brain (SMB) was performed. Subsequently, the pooled 5th SMB passage was inoculated intracerebrally (i.c.) in adult and suckling mice. The full genome sequences of the isolated 5th passage, in vivo and in vitro, were compared at both the consensus and the quasispecies level with the MSV. Additionally, the full genome sequence of the 6th SMB passage from the individual animals was determined and compared. Full-length integration of the double glycoprotein and modified base substitutions at amino acid position 194 and 333 of the glycoprotein could be verified in all 5th and 6th passage samples. Overall, 11 single nucleotide polymorphisms (SNPs) were detected in the 5th pooled SMB passage, 4 with frequency between 10 and 20%, and 7 with between 2.5 and 10%. SNPs that resulted in amino acid exchange were found in genes: N (one SNP), G (four SNPs), and L (three SNPs). However, none of these SNPs were associated with reversion to virulence since all adult mice inoculated *i.c.* with this material survived. In the individual samples of the 6th SMB passage 24 additional SNPs (>2.5%) were found, of which only 1 SNP (L-gene, position 6969) had a prevalence of >50% in 3 of 17 samples. The obtained results confirmed the stable expression of genetic modifications and the genetic stability of the consensus strain after serial in vivo and in vitro passaging.

## 1. Introduction

Rabies is a viral disease with one of the highest case fatality rates; every year an estimated 59,000 people succumb to this disease, and in most cases the virus is transmitted to humans by rabid dogs [1]. In Europe and North America, human rabies cases are relatively rare due to the absence of dog-mediated rabies. However, these continents cannot be considered rabies-free since the rabies virus persists in certain wildlife reservoir species. Oral vaccination has been developed into a highly successful method to eliminate rabies from the red fox population (*Vulpes vulpes*) in large areas of Europe and Canada [2,3]. First generation attenuated oral rabies virus vaccines, including SAD Bern, SAD B19, SAD P5/88, and ERA BHK21 contributed decisively to this achievement [4,5,6]. This method was subsequently adapted to target other wildlife rabies reservoir species, including raccoon dogs (*Nyctereutes procyonoides*), coyotes (*Canis latrans*), gray foxes (*Urocyon cinereoargenteus*), striped skunks (*Mephitis mephitis*), and raccoons (*Procyon lotor*) [2,7,8].

First generation oral rabies virus vaccine strains have been attenuated by conventional in vitro and in vivo passaging in order to restrain their replication in the targeted host so that the virus induces immunity rather than disease. The genotypic causation of the observed attenuation remains unidentified for these vaccine strains and experimental studies demonstrate that they have residual pathogenicity [9,10,11,12]. Furthermore, vaccine-associated rabies cases have been reported in areas where these vaccines have been distributed [5,13,14,15,16]. However, it is unclear whether these cases arose as a result of residual pathogenicity or reversion to virulence. The latter could be induced by back-mutation of attenuating mutations, compensatory mutations elsewhere in the genome, or changes in quasispecies diversity [17]. However, it has been concluded that most of these vaccine-associated cases arise due to host factors and not from the selection of a more virulent genotype within the quasispecies [6]. Furthermore, recent studies have shown that the most widely used 1st generation oral rabies virus vaccine strains, SAD Bern and SAD B19, consist of a mixture of variants [18,19]. As a result of the safety concerns associated with these vaccine strains, more attenuated oral rabies vaccine constructs were developed. Besides recombinant vaccinia virus and human adenovirus type 5 constructs expressing the rabies virus glycoprotein [8,20], 2nd generation oral rabies virus vaccines were developed by selecting highly attenuated mutants using monoclonal antibodies [21]. More recently, it became feasible to construct rabies viruses with targeted genomic modifications, thus, allowing the development of highly attenuated 3rd generation live-attenuated oral rabies virus vaccines that also contain one or more safeguards against reversion to virulence [22,23]. The use of the reverse genetics systems to introduce precisely defined mutations into viral genomes is, therefore, considered a milestone in veterinary medicine [24]. Using genetic engineering for vaccine development means genotype determines virus phenotype and not the other way around, as with traditional 1st generation attenuated vaccine viruses [17].

Recently, a 3rd generation oral rabies vaccine based on the SPBN GASGAS rabies virus strain received marketing authorization in the EU for oral vaccination of foxes and raccoon dogs against rabies, EMEA/V/C/004387 [25]. As part of the licensing procedure of replication competent veterinary vaccines, studies to examine reversion or increase in virulence by serial passaging of the candidate vaccine in the target species using the recommended route of administration is required; if justified, an alternative route of administration may be used. However, no in vitro propagation of isolated material between the passages is allowed [26,27]. As the vaccine construct is rapidly cleared from the site of entry after oral administration in the target species [28,29], investigating reversion to or increase in virulence by the recommended route of administration is not feasible. Even when the construct was inoculated directly into the brain of a potential target species, the small Indian mongoose (*Herpestes auropuntatus*), it did not induce disease and the vaccine virus was rapidly cleared from the inoculation site [30]. Furthermore, even with the less attenuated parenteral vaccine strain, SAD B19, it was not feasible to passage the virus serially in foxes after *i.c.* inoculation without in vitro propagation of the isolated material between the successive passages [9]. Hence, it is not feasible to test reversion to or increase in virulence by serial passaging by the recommended or alternative route of administration in the target species. As an alternative, the Pharm. Eur. recommends serial passaging of oral rabies vaccines in suckling mouse brains (SMB) [31]. Additionally, it is necessary to demonstrate the retention of the defined phenotypic properties and the genetic structure of the strain throughout seed lot production. Virus propagation during the manufacturing process can lead to the appearance and accumulation of mutants that may alter the efficacy and safety properties of the vaccine virus, especially in the case of RNA viruses with an inherently high mutation rate. Unfortunately, traditional sequencing methods are only able to detect the consensus sequence as an aggregate of all variants within the population and not individual variants occurring at low frequencies [32,33]. The development of next-generation sequencing (NGS), also known as high-throughput sequencing, has enabled less abundant mutations to be detected.

In this study, the genetic stability and purity of SPBN GASGAS have been investigated after serial passaging in vitro and in vivo studies using NGS. The results show that the MSV SPBN GASGAS is pure and is genetically stable after five serial passages, as the mutations were only detected at relatively low frequencies and did not change the consensus sequence. Importantly, no reversion of virulence was observed, and the genetic modifications introduced were stably expressed and had not altered.

## 2. Materials and Methods

### 2.1. Vaccine Virus

SPBN GASGAS is a highly attenuated derivate of the SAD B19 oral rabies virus vaccine and genetically modified by site-directed mutagenesis. The genome of SPBN GASGAS is approximately 13 Kb and comprises six open reading frames (ORF) in the order 3′-N-P-M-G-G-L-5′. It lacks the pseudogene (ϒ) and two amino acids of the glycoprotein have been changed at position 194 (AAT [Asn] → TCC [Ser]) and 333 (AGA [Arg] → GAG [Glu]) by replacing all three nucleotides at these positions [23]. Arginine (Arg) or Lysine (Lys) at amino acid position 333 of the glycoprotein is required for rabies virulence in adult mice [34,35], hence, a modification of this position renders the virus construct apathogenic for adult mice, even after direct *i.c.* inoculation. However, it was shown that a compensatory point mutation at position 194 of the glycoprotein partially restored virulence (pseudoreversion) after serial in vivo SMB passaging. Consequently, all three nucleotides at this amino acid position 194 were replaced and it was shown that this construct, SPBN GASGAS, was genetically stable and did not revert to virulence [36]. Finally, an additional glycoprotein gene containing the same genetic modifications was incorporated between the G and L gene [37]. The overexpression of the rabies virus glycoprotein in this construct, SPBN GASGAS, further increased safety by enhancement of apoptosis of infected cells and reduced virus replication in vivo [38].

### 2.2. In Vitro Studies

The in vitro studies for genetic stability were performed by serial passaging SPBN GASGAS five times on the adherent BHK21 Clone 13 production cell line. Cells from the working cell seed (WCS) were seeded with a cell number of 6 × 10^4^ cells/cm^2^ on 1-layer cell factories (CF1) using Minimal Essential Medium (MEM) with 5% fetal bovine serum (FBS) and 2% L-Glutamine as cell growth medium. The cells were grown for 48 h, afterwards a complete medium change was performed using a Glasgow Minimal Essential Medium (GMEM) with 5% FBS and 2% L-Glutamine. The medium volume used per CF1 was 180 mL. Virus infection was conducted using a multiplicity of infection (MOI) of 0.02 and the infected cells were incubated for 7 days. Afterwards, the virus containing supernatant was harvested and frozen at −80 °C until the next passaging round. This procedure completely mimics the routine large scale production process (which routinely uses 10-layer cell factories CF10) as a scale-down model and the passage limits that may occur in production.

### 2.3. In Vivo Studies

The in vivo genetic stability was tested by serial passages in NMRI suckling mice (Charles River GmbH, Sultzfeld, Germany). Prior to inoculation, five suckling mice (between 6 and 24 h old) were selected per litter, the other litter mates were removed from the cage. Fifteen suckling mice received 0.01 mL MSV material with a titer of 10^7.6^ foci forming unit (FFU)/mL by the *i.c.* route. Mice that died before the 5th day after inoculation were not used for re-isolation of virus material. On the 5th day, the mice were euthanized, the brains removed, pooled, and homogenized in MEM. Subsequently, the homogenate was clarified by centrifugation and filtration (syringe filters: glass fiber filter—1 μm and cellulose acetate—0.45 μm) and then titrated. This first passaged virus material was inoculated into the next group of fifteen suckling mice. This procedure was repeated until the 4th passage was isolated. This material was inoculated into thirty instead of fifteen animals to ensure that sufficient brain material of the 5th passage was available. The complete genome sequence of the pooled 5th passage SMB isolate was investigated by NGS for purity. The phenotypic characteristic of this isolate was also investigated by *i.c.* administration (0.03 mL) in sixteen adult NMRI-mice that were observed for 28 days. The administration of the vaccine in adult mice was carried out under isoflurane narcosis (CP Pharma, Burgdorf, Germany). Finally, the pooled 5th passage SMB isolate was administered *i.c.* to suckling mice and the isolated 6th passage from the individual animals was examined by NGS.

### 2.4. Assay

#### 2.4.1. Virus Titration

The titer of the samples was determined in tissue culture using the rapid fluorescent antigen test. For this purpose, wells of a 96-well plate were filled with 0.1 mL of BHK 21 Clone 13 cell suspension in MEM mixed with 20% newborn calf serum (NCS). Subsequently, the wells were infected with 0.1 mL of a 10-fold serial dilution of the isolated virus, except for the wells used for positive and negative controls. After 48 h of incubation (35 °C, 4–5% CO_2_), supernatants were removed, and the cells were fixed with 80% acetone at room temperature for 30 min. Wells were rinsed with phosphate buffered saline (PBS) and air-dried. Fluorescein isothiocyanate (FITC)-labeled monoclonal anti-rabies IgG (Fujirebio Germany GmbH, Hannover) was added and wells were stained for 30 min at 37 °C. Finally, the number of fluorescent foci were counted by fluorescent microscopy. The titer was expressed in FFU per mL.

#### 2.4.2. Next Generation Sequencing (NGS)

The original MSV, the different mouse brain samples (the pooled 5th passage SMB isolate and the isolated 6th passage from individual animals), and the MSV cultivated over five in vitro passages were sequenced and compared.

During the investigation of the genetic stability of SPBN GASGAS, the sample preparation for sequencing was optimized in parallel. This process had no influence on the sequencing result but increased the output on sequencing data. This enabled fewer sequencing runs and the possibility to sequence more samples in parallel. The final and most efficient preparation method is described here.

#### 2.4.3. Total Nucleic Acid (TNA) Extraction

165 µL samples were digested with 1/10 volume 0.2 U/µL micrococcal nuclease (Sigma-Aldrich, Darmstadt, Germany) and 1/100 volume 0.1 M CaCl_2_-solution (Sigma-Aldrich) at 28 °C for 30 min at 700 rpm to reduce the residual host cell DNA and RNA. The reaction was stopped with 1/10 volume 0.5 M EDTA pH 8.0 (Lonza, Basel, Switzerland).

The extraction of TNA was performed with the final volume after nuclease digestion with the MasterPure Complete DNA and RNA Purification Kit (Lucigen, Middleton, WI, USA). An up-scaled protocol of ‘Lysis of Fluid or Tissue Samples’ for a 200 µL sample volume was used. The TNA was resuspended into 40 µL of molecular biology water (Lonza).

#### 2.4.4. cDNA Synthesis

cDNA synthesis from 5 µL of TNA was performed using the Maxima H Minus First Strand cDNA Synthesis Kit (Thermo Scientific, Vilnius, Lithuania). At variance with the kit guidelines, priming with 1 µL of Random Hexamer Primer, 1 µL of 10 mM dNTP Mix and 8 µL of molecular biology water (Lonza) was carried out prior to the denaturation step. The synthesis reaction was performed at 55 °C.

#### 2.4.5. Specific Amplification

The target region was enriched using five different primer pairs resulting in five fragments between 2408 bp and 2948 bp (Table 1). Primer sites were chosen so that all gene coding regions of SPBN GASGAS were presented. Primers were located both in non-coding and coding regions, thus, generating overlapping fragments to prevent sequencing errors arising due to failed primer oligo synthesis (Figure 1). The amplification was performed with 3 µL of cDNA, 2 µL of each primer pair (each primer 5 µM), 7.5 µL of molecular biology water (Lonza) and Phusion Flash High-Fidelity PCR Master Mix (Thermo Scientific) to a total volume of 25 µL. The initial denaturation was performed as follows: 20 s at 98 °C; 38 amplification cycles with 5 s denaturation at 98 °C; 5 s annealing at 60 °C; 50 s elongation at 72 °C. Final elongation was performed for 1 min at 72 °C and for an extended time at 10 °C.

#### 2.4.6. Gel Electrophoresis and Gel Extraction

25 µL of PCR product were separated in 1.2% agarose gels. The gels were stained with ethidium bromide before pictures were taken under UV light. The amplicons of interest were extracted with the Wizard SV Gel and PCR Clean-Up System (Promega, Madison, WI, USA). The amplicons were eluted with 40 µL of molecular biology water (Lonza).

#### 2.4.7. DNA Quantification and Sequencing

The DNA quantification was performed with the Quant-iT PicoGreen dsDNA Assay Kit (Life Technology, Carlsbad, CA, USA) using 2.5 µL of sample in 200 µL of reaction mixture. Fluorescence was measured with the Multimode reader infinite M1000 PRO (Tecan, Nänikon, Switzerland). The Nextera XT Library Prep Kit was used to prepare the library for sequencing. The sequencing reaction was performed with the MiSeq Reagent Kit v3 600 Cycles (paired end reads, read length reduced to 250 bp + 250 bp to get high quality reads) for mouse brain samples and with the MiSeq Reagent Kit v3 150 Cycles (paired end reads, 75 bp + 75 bp) for MSV and MSV + 5 samples on a MiSeq sequencer (all Illumina). All amplicons were pooled into one sequencing library.

#### 2.4.8. Sequencing Data Analysis

The overall Q-scores of read per position were >20 and, thus, no quality filtering was performed. The sequencing reads for each sample data set were mapped onto the amplified regions of the reference genome using different software. The initial alignment was conducted with DNASTAR Lasergene SeqMan assembler suite (version 11.2.1 and 13.0.0, respectively). For further and more detailed data analysis, the following bioinformatical strategy was pursued.

All programs were installed with Conda v4.9.0. The quality of fastq files was evaluated with FASTQC software (0.11.9). Reads were aligned to the SPBN GASGAS reference sequence (Genbank: MH660455) with BWA MEM v0.7.17 [39] using default options. Read depth per genomic position was computed with SAMtools v1.10 [40]. SNP analysis was performed with Bcftools v1.10.2 using mpileup function [41]. The genomic positions, for which a median of read depth was lower than 3000 in at least one of the samples in a comparison were excluded and a SNP frequency threshold of 2.5% was implemented. SNP frequencies below 2.5% were shown if the comparator sample had a SNP frequency of 2.5% or higher (otherwise a comparison would not be possible).

Generation of consensus sequence and haplotype analysis for each sample were performed with Cliquesnv v2.0.2 using default options for SAM files [42]. Consensus and haplotype sequences were compared to the reference sequence by multiple alignment with Clustal Omega v1.2.4 [43].

SPBN GASGAS genome structure and binding sites of primers used to generate amplicons were visualized using DNA Features Viewer v3.0.3 [44].

#### 2.4.9. Ethics Statement

The in vivo study was performed under the principles of the VICH guideline for Good Clinical Practice (VICH GL9) and conducted under permit 42502-3-658 IDT issued by the appropriate authorities of the federal state of Saxony-Anhalt in Germany.

## 3. Results

### 3.1. In Vitro

The sequencing verified the integration of both glycoprotein genes and the targeted base substitutions at position 194 and 333 in both G genes. Overall, no variant affecting the SPBN GASGAS consensus sequence (frequency > 50%) was detected in the MSV or after five passages in adherent cell culture (Table 2). Seven minor variants of frequency below 6% were present at the beginning in MSV. After five passages, the frequency of most variants decreased, and for one SNP, an increase from 0.44% to 2.75% was observed (position 9748: G to T) (Table 2). This nucleotide is located on the L gene and led to a nonpolar to nonpolar amino acid exchange, from leucine (L) to phenylalanine (F). The low frequency of variants in MSV along with the fact that all variants ≥ 2.5% in MSV decreased after five passages indicates that the virus sequence was stable in vitro after serial passaging. The titer of the 4th and 5th in vitro passage was determined and found to be identical to the titer of MSV (Table 3), indicating that no change in virus replication capacity occurred.

### 3.2. In Vivo

Similar to the in vitro analysis, the sequencing verified the integration of both glycoprotein genes and base substitutions at position 194 and 333 in both G genes in the pooled 5th and individual 6th SMB isolates. The titers of the pooled SMB passages are shown in Table 3.

High-throughput sequencing of amplicons generated for samples was performed after the 5th (P5, pooled sample of 29 mice) and 6th (P6, 17 individuals) in vivo passages. Moreover, between 97.44 and 99.51% of sequencing reads across all samples were aligned to the reference genome of SPBN GASGAS, implying efficient amplicon purification and no sequencing of untargeted regions (Figure 1a). A high coverage of the reference genome was observed, with a median range from 33,128–57,070 alignments across all samples (Figure 1b), except for approximately 20 nucleotide regions of primer binding sites. This very deep coverage for all alignments provided a high confidence for consensus sequence recovery and variant information (Figure 1c). No difference between a consensus sequence recovered from P5 and the SPBN GASGAS MSV reference sequence was observed. In P6, changes in consensus sequence occurred only in 4 out of 17 mice and just one nucleotide exchange in whole genome sequence per mouse was found (Table 4). In mouse M1K12, the exchange appeared in genome position 557 (gene N, A to G substitution), whereas in 3 mice (M1K9, M4K9, M2K9) the exchange appeared in position 6969 (gene L, A to G substitution). Both these exchanges were nonsynonymous, leading to the replacement of a polar amino acid to another polar amino acid (threonine to alanine and asparagine to serine).

The genetic stability of SPBN GASGAS was further evaluated by variant calling without performing base-call quality scores correction, thus, potentially allowing false positive variant calls. It should be noted that the amplicons of G1 and G2 genes were pooled prior to sequencing, hence, the reads generated for the G1 gene could be aligned to the G2 gene sequence and vice versa. The ambiguous reads after mapping were not removed, thus, the presented results on SNP frequencies in these positions (genomic positions of G1: 3315–4889 and G2: 4956–6530 in Figure 1c) represent the worst-case scenario for a single G gene. In total, 11 SNPs were found in the P5 pooled sample, with frequencies ranging from 2.52% to 20.54%. The three highest frequency SNPs were found in genes N and L (positions 557, 6836 and 8692; Figure 1c). Overall, in the isolated P6 sample, between 14 and 25 SNPs per sample of the individual animals could be detected (Figure 1c). Importantly, no SNP was found in the positions where site-directed mutagenesis was purposely introduced into the SPBN GASGAS genome, both in the P5 and P6 samples (Figure 1c, sites indicated with “×”), which implies that the modifications of virus strain remain resistant to genetic changes over in vivo passages.

Changes in SNP frequencies in P5 and P6 samples were evaluated to infer whether variants accumulate or if their frequencies tend to increase over passages. Changes in SNPs occurring in G genes from the P5 to P6 passage are shown in Figure 2. Changes in SNPs for the remaining genome (N, P, M, and L genes) from the P5 to P6 passage are shown in Figure 3. New SNPs compared to the MSV consensus sequence appeared in 24 genomic positions (Table 5). One new SNP occurred in 13 samples of P6 (position 1564 in the G2 gene), and this nucleotide change may lead to the substitution of histidine ‘H’ to glycine ‘G’ (a polar to non-polar amino acid exchange). Ten SNPs, which occurred in P5, were also found in all samples of P6 (category ‘stable’), whereas in genomic position 7609, a SNP of low frequency (2.52% to 2.59%) was present in P5 and five samples of P6 (category ‘stable’). These common SNPs of P5 and P6 were rather stable or increased in P6 compared to their frequency in P5 (Figure 2 and Figure 3). The most dramatic increases in the SNP frequencies, present in more than 50% of P6 samples, were observed at: two positions in the G gene, one in the N gene, and two in the L gene. Notably, at nucleotide position 877 of the G gene, the average SNP frequency increase was 12.19% (±6.38%) in ten samples. Furthermore, at nucleotide position 1546 of the G gene, there was a 14.95% (±9.79%) average increase of SNP frequency in 14 samples. Overall, this suggests that for in vivo passages, the SNPs accumulated in P5 persist in P6, and that new SNPs emerge randomly in different samples. As the P5 samples consisted of a virus pool from different animals, it is not possible to conclude whether SNPs that were “inherited” in P6 from P5 developed in parallel in many mice or represent the most frequent allele originating from single or few animals.

All adult mice inoculated i.c. with the pooled P5 sample remained healthy during the entire observation period, indicating no (partial) reversion to virulence.

## 4. Discussion

Since the rabies virus belongs to the genus *Lyssavirus* of RNA viruses in the family *Rhabdoviridae*, mutations are expected to emerge rapidly due to the lack of proofreading activity of RNA virus polymerases [45]. It is generally accepted that the average mutation rate is from 10^−4^ to 10^−5^ mutations per nucleotide per round of RNA replication, although for another rhabdovirus, vesicular stomatitis virus (VSV), a ten-fold higher mutation rate was reported [46]. Consequently, rabies virus populations—including vaccines—normally consist of a widely dispersed mutant distribution, rather than a homogeneous one formed by a single most-fit sequence. Although most mutations that emerge are not harmful to vaccine properties, the consequences of their accumulation must be identified to ensure vaccine safety and potency.

In this study, it was shown that the vaccine strain, SPBN GASGAS, was genetically stable, including the incorporated targeted genetic modifications, after five serial passages in SMB and especially in cell culture lines. After five serial passages in the manufacturing cell line, only one SNP was detected with an abundancy of 2.75%. Of the eleven SNPs detected in the pooled 5th SMB passage, none had a relative abundancy over 50% and only four SNPs were found with a relative frequency between 15% and 20%. The pooled 5th SMB passage was inoculated in adult mice (i.c.) and none of the animals succumbed to rabies, indicating that none of the observed mutations had virulent properties.

In the individual isolates from the 6th SMB passage, twenty-four additional SNPs (>2.5%) were identified. Thirteen of these twenty-four SNPs were detected in non-coding regions or non-coding primer regions. It is unlikely that the additional mutations observed in the 6th SMB passage would cause a reversion to a more virulent phenotype, as none of the observed amino acid changes were found in any of the known pathogenicity determining sites as compiled by Eggerbauer et al. [47]. The analysis was based on short read data allowing inference of possible haplotypes but not determination of true haplotypes. Thus, it is unknown if multiple mutations occurred in the genome of a single virion or whether the observed mutations occurred on multiple virus particles.

It should be noted that base-calling errors are inherent to any sequencing technology. Thus, SNPs at low frequencies are often difficult to distinguish from technical errors. A high absolute read depth of sequencing and a high number of templates in a sample do not guarantee that a viral population is accurately reproduced [48]. For brain tissue samples with often relatively low levels of viral population, an amplification step is usually required, and this can also produce mutations [49]. Additionally, in these studies, samples were amplified before analysis and, therefore, the results may be biased since prior amplification may result in biased data due to the non-random selection of viral variants, especially when the threshold is set too low [50]. In the literature, a wide range of cut-offs is used—from 0.1–10% [19,33,51,52,53]. For the Illumina sequencing platforms used in this study combined with workup (quality of RT Polymerase enrichment and purification of rare variants), variant thresholds tend to fall in the range from 0.5% to 3% [53].

In this study, a weakness in the sequencing strategy was to pool amplicons of SPBN GASGAS, which meant that the origin of the sequencing reads could not be discriminated between the two copies of the G gene. A separation of G1 and G2 amplification products into different sequencing libraries, or generation of longer sequencing reads with other sequencing technology, would provide more insight into variability of G1 and G2 genes.

It should be noted that the requested serial passaging in SMB to test genetic stability is highly artificial for a rabies virus vaccine offered in a bait and the observed mutants may not have appeared under natural conditions. For example, Leonard et al. [53] showed that the bottleneck stringency was greatly influenced by the route of administration. For oral vaccination, the vaccine virus is released in the oral cavity of the host and taken up predominantly by the palatine tonsils, where restricted replication takes place followed by rapid clearance [54]. The uptake by tonsils is a first bottleneck; only a limited amount of the virus particles released from the blister will enter the host [28,54]. It is assumed that SPBN GASGAS replicates relatively slowly due to the genetic modifications inserted compared to its parental vaccine strain, SAD B19, thus, the virus population remains relatively small with a low number of replication cycles before clearance by the immune response. Consequently, the accumulation of mutations would, therefore, be limited compared to viruses that show extensive replication and spread within the host after direct *i.c.* inoculation. Therefore, to investigate genetic stability of the vaccine during the actual manufacturing process is more meaningful. The vaccine strain has been adapted to cell culture and, therefore, it can be expected that fewer mutations will occur in this substrate than in SMB, a completely new environment for the virus strain. Therefore, the genetic stability of the construct was also investigated after five serial passages in the production cell line, BHK Cl13, to examine whether the selection pressure is different from required serial SMB passaging. The results showed that the number of low frequency mutants was lower after in vitro than in vivo passaging. For field isolates, it is the other way around; these isolates cultured in BHK cells had more SNPs than tissue samples [51]. Viral genomic sequences sampled early (low passage number) on new hosts will contain an excess of mildly deleterious variants that would eventually be eliminated by purifying selection. Furthermore, initially purifying selection may be relaxed in new host systems (SMB). Both these scenarios would increase the number of SNPs observed and ‘inflate’ genetic instability [55]. Taken together, most substitutions observed after in vivo passaging are not a result of positive selection after serial passaging, but rather the result of genetic drift. As described previously, the narrow bottleneck due to the route of administration of oral rabies vaccines reduces the amount of transferred viral genetic diversity and, thus, may slow the rate of possible viral adaptation within the host.

The ability to analyse vaccine viruses at such a detailed level is challenging both for licensing authorities and vaccine producers to develop new quality control criteria. NGS may have produced a revolution and opened new perspectives for research and diagnostic applications; however, there are still some unresolved issues regarding how to utilize this innovative method for regulatory affairs. Consequently, the current definition of a viral vaccine strain as a uniform entity needs to be reconsidered [19].

## 5. Conclusions

The study revealed a stable consensus sequence for SPBN GASGAS over multiple passages in cell culture and in vivo, including the incorporated genetic modifications. To guarantee safety of the vaccine virus, every batch will be examined for purity and identity as part of quality control. This study, however, underscores the need for updated guidelines for evaluating genetic stability of live attenuated vaccines.

## Figures and Tables

**Figure 1 viruses-14-02136-f001:**
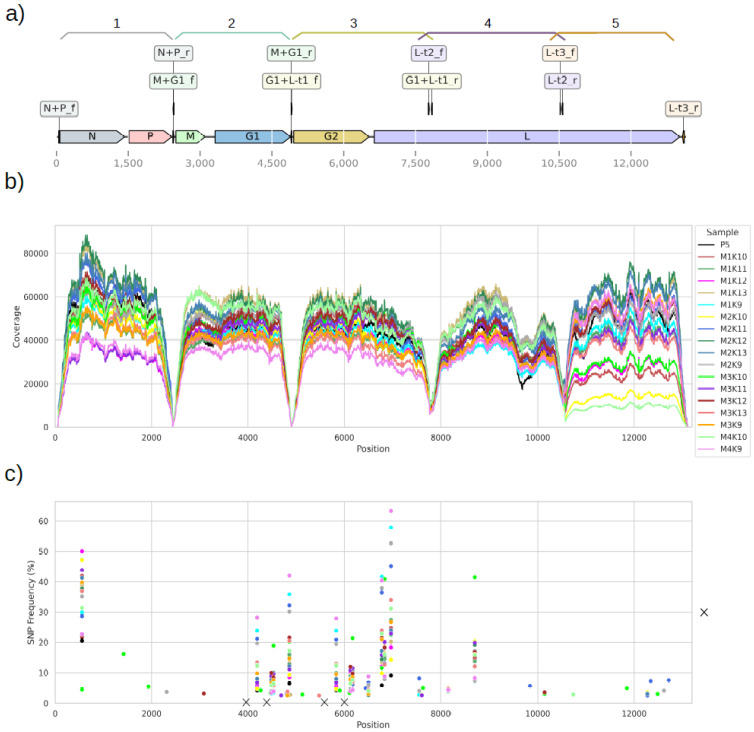
Genomic location of amplicons and variants. (**a**) Genome sequence of Rabies lyssavirus SPBN GASGAS strain and primer pairs used to generate amplicons (f—forward, r—reverse primer). The amplicons generated are indicated above according to numbering in Table 1. (**b**) Whole-genome read coverage for pooled 5th SMB passage (P5) sample and 17 SMB samples from passage 6 (MxKx—animal number). (**c**) SNP frequency in P5 and 17 individual SMB samples from passage 6. Positions, in which a change in consensus sequence occurred in at least one sample, are indicated above the given position together with the nucleotide and amino acid changes. The mutation introduced in SPBN GAGSAS strain, namely AAT [Asn] → TCC [Ser] (aa position 194) and AGA [Arg] → GAG [Glu] (aa position 333) are indicated by highlighting the middle nucleotide with “×”.

**Figure 2 viruses-14-02136-f002:**
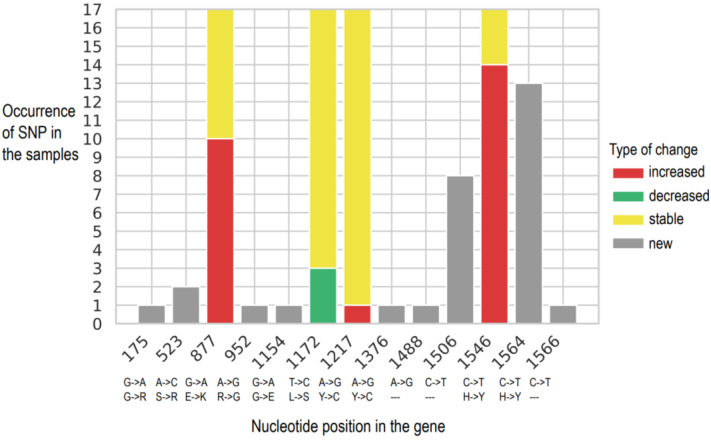
Changes in SNPs occurring in G genes from P5 to P6 passage. Nucleotide position in the glycoprotein gene—not genome—is indicated on X axis. Frequency of each SNP in samples from P6 was compared to P5 as basis, i.e., increased: P6 frequency > P5 frequency + 5% in P5, stable (±5% change in frequency), decreased: P6 frequency < P5 frequency—5%, new: SNP occurring just in P6 and not in P5. Nucleotide exchange and the possible consequence at the amino acid level is indicated below the graph (“---” indicates no amino acid exchange—silent mutation).

**Figure 3 viruses-14-02136-f003:**
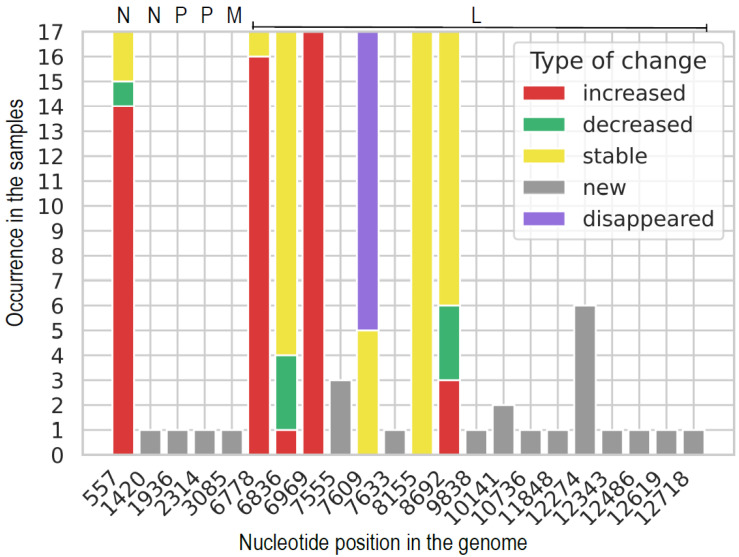
Changes in SNPs occurring in N, P, M, and L genes of SPBN GASGAS genome from P5 to P6 passage. Genomic position is indicated on X axis. Frequency of each SNP in samples from P6 was compared to P5 as basis, i.e., increased: P6 frequency > P5 frequency + 5% in P5, stable (±5% change in frequency), decreased: P6 frequency < P5 frequency—5%, new: SNP occurring just in P6.

**Table 1 viruses-14-02136-t001:** Primer sequences for gene amplification.

	Primer	Nucleotide Sequence	Resulting Fragment Size
1	GASGAS_N+P_f	GCAAAAATGTAACACCCCTA	2408 bp
	GASGAS_N+P_r	GGGACATCTCGGATTTTATT	
2	GASGAS_M+G1_f	ACAATAAAATCCGAGATGTCC	2482 bp
	GASGAS_M+G1_r	GATCGTTGAAAGGACGTTAAT	
3	GASGAS_G1+L-t1_f	CCTTTCAACGATCCAAGTC	2948 bp
	GASGAS_G1+L-t1_r	CTTGCTAAGCACTCCTGGTA	
4	GASGAS_L-t2_f	TATCGAAAGGGTCTGTCAAA	2811 bp
	GASGAS_L-t2_r	GTGCCAATGAAACGTAGAGT	
5	GASGAS_L-t3_f	CACAGACATGGACATCAGAG	2605 bp
	GASGAS_L-t3_r	ATCAAACAACCAAAGGTTCA	

**Table 2 viruses-14-02136-t002:** Genomic location and frequency of variants in MSV and MSV after 5th passage (MSV + 5P) in adherent cell culture. For MSV + 5P, the SNP frequencies below 2.5% are presented for the comparison of variant evolution.

Nucleotide Position	Reference Nucleotide	Alternative Nucleotide	Amino Acid Change	Gene	SNP Frequency (%)	
					MSV	MSV + 5P
514	T	C	-	N	4.49	0.04
517	G	A	-	N	4.78	0.06
693	G	A	G-->E	N	4.5	0.04
1241	G	A	D-->N	N	5.14	0.06
1750	G	T	E-->D	P	4.12	0.05
9748	G	T	L-->F	L	0.44	2.75
12,454	T	C	-	L	3.7	1.20

**Table 3 viruses-14-02136-t003:** Titer (^10^log FFU/mL) of the original material (MSV) and the different serial in vivo (SMB) and in vitro (BHK21 Clone 13 cells) passages; the number of SMBs used per passage is listed in parenthesis; n.d.—not determined.

Passage	In Vivo	In Vitro
MSV	7.6	
1st	7.1 (15)	n.d.
2nd	8.0 (11)	n.d.
3rd	8.3 (10)	n.d.
4th	7.4 (9)	7.6
5th	7.3 (29)	7.6

**Table 4 viruses-14-02136-t004:** Sequencing summary statistics in samples from the pooled 5th (P5) and individual 6th passage (MxKx) SMB material.

Sample	Total Reads per Full Genome	% of Mapped Reads	Median of Reference Coverage (Reads)	Genomic Position of Changes in Consensus Sequence	Number of SNPs with >2.5% Frequency
P5	2,526,454	99.46	43,037	none	11
M1K10	2,337,358	99.32	44,212	none	11
M1K11	2,148,128	97.59	39,094	none	11
M1K12	1,986,546	99.23	33,128	557	12
M1K13	3,180,882	99.24	57,070	none	14
M1K9	2,248,544	98.99	41,230	6969	11
M2K10	2,021,776	99.33	37,857	none	13
M2K11	2,824,706	99.01	47,369	none	17
M2K12	3,190,578	99.27	55,116	none	13
M2K13	3,025,394	99.11	52,629	none	11
M2K9	2,771,734	99.08	49,271	6969	14
M3K10	2,341,254	99.30	43,237	none	19
M3K11	2,225,710	98.38	40,057	none	13
M3K12	2,614,868	99.20	47,737	none	11
M3K13	2,065,018	99.05	37,609	none	12
M3K9	2,352,068	97.44	40,822	none	11
M4K10	2,389,962	99.51	46,726	none	11
M4K9	2,074,752	97.81	34,253	6969	10

**Table 5 viruses-14-02136-t005:** List of the 24 new SNPs in 6th SMB passage shown in Figure 2 and Figure 3 as gray boxes, in Figure 3 for G genes, both genomic positions are indicated.

Position	Reference	Alternative	Amino Acid Change	Gene
1420	A	G	-	N
1936	A	C	-	P
2314	T	C	-	P
3085	C	T	S-->F	M
3489/5130	G	A	G-->R	G
3837/5478	A	C	S-->R	G
4266/5907	A	G	R-->G	G
4468/6109	G	A	G-->E	G
4690/6331	A	G	Y-->C	G
4802/6443	A	G	-	G
4820/6461	C	T	-	G
4860/6501	C	T	H-->Y	G
4880/6521	C	T	-	G
7555	G	A	-	L
7633	C	A	D-->E	L
9838	A	G	-	L
10,141	G	A	-	L
10,736	G	A	D-->N	L
11,848	C	A	-	L
12,274	C	T	-	L
12,343	C	T	-	L
12,486	A	C	K-->T	L
12,619	G	A	-	L
12,718	C	T	-	L

## Data Availability

The original data can be provided upon reasonable request and should be directed to the corresponding author.

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
