# Peer review of "Oral Rabies Vaccine Strain SPBN GASGAS: Genetic Stability after Serial In Vitro and In Vivo Passaging"

_viruses, 2022, doi:10.3390/v14102136_

Round 1

Reviewer 1 Report

In this study, genetic stability of a genetically modified, oral rabies vaccine strain, SPBN GASGAS, was examined by serial passages in both cultured cells and suckling mouse brains and the following next-generation sequencing (NGS) analysis. The results indicated that the passaged viruses retained Ser and Glu at positions 194 and 333 in the G protein, which are responsible for attenuation phenotype of the SPBN GASGAS. Also, the examination identified a large number of single nucleotide polymorphisms (SNPs) in the passaged strains, none of which was located in previously reported pathogenicity-determining sites. Based on these results, the authors concluded that the SPBN GASGAS has a high level of genetic stability in vitro and in vivo, thus suggesting that the strain stably maintain favorable phenotypes as an oral rabies vaccine strain in the manufacturing process and in the field, respectively. 

This study is of great scientific importance, examining the genetic stability in vitro and in vivo of the SPBN GASGAS strain used as an oral rabies vaccine for foxes and raccoons, which has been approved for marketing in the EU. However, I had an impression that the experimental design does not completely match a goal of this study. Specifically, the goal of this study is to confirm that the SPBN GASGAS does not acquire undesirable phenotypes including high virulence in the manufacturing process and in the body of an inoculated animal. Whereas the authors comprehensively analyzed genetic properties of the strain serially passaged in vitro and in vivo, they did not check phenotypes of the passaged viruses enough, thus not being able to discuss relation between genetic and phenotypic changes of the passaged viruses. Although I agree that the SNPs found in this study could not significantly affect the phenotypes of the passaged viruses, I believe that addition of phenotypic data would support the authors’ conclusions better.  

Specific comments

1)    As mentioned above, addition of experimental data showing phenotypes of the in vitro and in vivo passaged viruses would significantly improve this paper. More specifically, it is important to confirm that growth curve in BHK-21 cells and pathogenicity in adult mice of the SPBN GASGAS did not drastically change by serial passage in BHK-21 cells, in order to demonstrate that the SNPs identified in the passaged viruses do not affect productivity and safety of the oral vaccine. Also, it is better to confirm that avirulent phenotype of the strain in mice did not change after the serial passage in suckling mouse brains, to show that the SNPs do not increase its risk of reversion to virulence in the body. I know that the authors conducted the latter experiment but did not show the results in Results section. Such phenotypic data would support the genetic data shown in the current version.

2)    Throughout the manuscript, in vitro and in vivo passaging experiments are treated in the same breath. However, the former aimed to check genetic stability of the SPBN GASGAS in the manufacturing process, while the latter aimed to check that in the inoculated animal body. Thus, the data from the respective experiments should be described and discussed separately, to support better understanding by readers.

3)    In Figure 2, why did the read coverage in fragment 5 vary more clearly among samples, compared to those in other fragments?

4)    In Figure 3, two series of bars exist in parallel in the Z axis, but there is no explanation.  The authors need to specify which line represents what data.

5)    The legend of Figure 3 (lines 348-349) says “Nucleotide change and the possible consequence at the amino acid level is indicated below the graph”, but such indications are not found in the graph. 

6)    Paragraphs in Discussion section are not well organized. Please rewrite each paragraph so that one paragraph provides basically one message.

7)    Discussion (L374-376). The detailed data of the i.c. inoculation should be provided in Figures/Tables in Results section. Also, I am curious about virulence of each virus at the 6th passage in suckling mice, because we cannot deny the possibility that some SNPs could affect viral pathogenicity. Suckling mice can provide a better model when we check slight change in viral virulence.

8)    Discussion (L409-420). This paragraph is very confusing. The authors’ speculations are written like a proved theory, although the speculations are not supported by experimental data.

Conclusions (L444-446). I cannot agree with the idea that “NGS of vaccine material serially passaged in cell culture could replace the study in suckling mice…”. I believe that data obtained in this study are not sufficient to support the idea. Further accumulation and comparison of data from serious passages of various vaccine strains in cultured cells and suckling mice will be necessary to test the idea in the future. I agree that serial passage in suckling mouse brains is very artificial and does not provide a perfect model to examine risk of “reversion to virulence” in the animals after oral vaccination. This method can overestimate the risk, as it provides very strict conditions. However, “too much” is better than “too little” in this case, at least for now. 

Reviewer 2 Report

Major:

- Line 152-153: "Mice that died  before the 5th day after inoculation were not used for re-isolation of virus material." What are the cause of death for these mice? How often did it happen? The focus of the study is to characterize if reversion of the virus is observed. These dying mice could be an indication of a pathogenic reversion of the virus during serial passage. Could you please provide Keppler-Meyer curves of mice survival between injection and euthanization/havest, and comment as to the nature of these deaths?

- Line 154: Why did the authors choose to pool brain homogenates for every serial passage rather than keeping different "lineages". Wouldn't individual serial passage of those provide a higher degree of coverage for SNPs emergence?

- What is the difference between figure 1 and figure 2A? Figure 2A is nice to visualized the coverage and SNPs position and should be conserved. Figure 1 is not necessary and authors can referenced the figure 2A instead.

- Figure 2C: It would be beneficial if Passage 5 SNPs frequency was plotted in a much more different color than any of the 17 P6 mice to ease visualization.

- Statement lines 292-293 should be supported by a reference.

- Figure 3: What is the difference between the 2 rows of bar graphs? Does it need to be in "3D"? image quality is very low and barely readable.  Overall, that figure should be simplified (e.g. 2D layout) and clearly defined.

Minor, Typos and English:

Although the manuscript is readable and comprehensible, there are multiple typos and English inaccuracies such as (but not limited to) below. Please have another round of editing.

- Abstract (line 36-38): Authors might need to rephrase last sentence.

- Line 92: "Already" should be replaced.

- Line 98: "Also" should be replaced.

- Line 262: Extra "." to be removed after (L).

- Missing "." at the end of sentences (lines 264,266,273,...)

Round 2

Reviewer 1 Report

Unfortunately, the revised manuscript is not improved significantly: the essential problems I pointed out previously have not been addressed. I would like to emphasize again that the genetic data shown in this paper are not so valuable without phenotypic data. 

Please see my comments shown below for details.

1)  Although the authors argued that the submitted manuscript must be seen from a regulatory view and not from a scientific perspective in their comments, I cannot agree with it. You should remember that Viruses is a scientific journal.

2)  I pointed out that the data did not support their conclusion that NGS of vaccine material serially passaged in cell culture could replace the study in suckling mice, but it appears from the revised manuscript that the authors did not change the conclusion. Their comments on this point are also not convincing.

3)  Although the authors moved the statement about virulence of SMB p5 virus in adult mice to Results section (Lines 354-355), this modification is not satisfactory. The authors should show the data in graph(s) or table(s) objectively indicating that the SMB p5 virus remained attenuated even after serial passages in SMB. I strongly recommend that the authors should compare body weight curves of mice inoculated intracerebrally with the SPBN GASGAS and the SMB p5 virus.

4)  The authors insisted in their comments that checking virulence of all the SMB p6 viruses in adult mice is not feasible. I agree with this comment. However, it is feasible to pick several representative from the SMB p6 viruses and to check their virulence in adult mice. I still believe that addition of such data will significantly improve the manuscript. 

Author Response

Unfortunately, the revised manuscript is not improved significantly: the essential problems I pointed out previously have not been addressed. I would like to emphasize again that the genetic data shown in this paper are not so valuable without phenotypic data. 

Please see my comments shown below for details.

1)  Although the authors argued that the submitted manuscript must be seen from a regulatory view and not from a scientific perspective in their comments, I cannot agree with it. You should remember that Viruses is a scientific journal.

 Reply: we do agree to this statement but as scientists we need to be realistic what is feasible and what is not. Comparing the workload on testing genetic stability in the frame of vaccine development, we went way beyond what is requested, even for human vaccines. From a regulatory perspective, you do your 5 passages in SMB and check if the targeted modifications applied are still present. If we would have done only this, we would never have submitted it for publication. However, we added the serial passaging in the production cell line, we added not only NGS of the pooled 5th passage both in vivo and in vitro, but also the individual NGS sequence of all 6th SMB passage material. Finally, we checked the phenotypic type (in terms of virulence) of the 5th SMB passage in adult mice. We considered the extra work as relevant to share with the scientific community and thus we submitted this manuscript

2)  I pointed out that the data did not support their conclusion that NGS of vaccine material serially passaged in cell culture could replace the study in suckling mice, but it appears from the revised manuscript that the authors did not change the conclusion. Their comments on this point are also not convincing.

 Reply: we accept this comment and adapted the text accordingly

3)  Although the authors moved the statement about virulence of SMB p5 virus in adult mice to Results section (Lines 354-355), this modification is not satisfactory. The authors should show the data in graph(s) or table(s) objectively indicating that the SMB p5 virus remained attenuated even after serial passages in SMB. I strongly recommend that the authors should compare body weight curves of mice inoculated intracerebrally with the SPBN GASGAS and the SMB p5 virus.

 Reply: Suckling mice are very sensitive to stress esp. after having received the test substance by i.c. inoculation. Hence, weighting such animals on a daily base could result in stress-induced mortality. Furthermore, the mother animal is very sensitive at this stage for disturbance and the risk that she would kill (and consume) her young cannot be ignored. This would not only jeopardize our study but weighting suckling mice is not allowed from an animal welfare standpoint. The authorities would never allow such a procedure when they evaluate the study protocol for approval

4)  The authors insisted in their comments that checking virulence of all the SMB p6 viruses in adult mice is not feasible. I agree with this comment. However, it is feasible to pick several representative from the SMB p6 viruses and to check their virulence in adult mice. I still believe that addition of such data will significantly improve the manuscript. 

Reply: yes, it would definitely be interesting. However, it remains rather arbitrary which isolates we should use as we do not have any indication which ones would be the most interesting. Furthermore, the artificial serial passaging of an oral rabies vaccine in the brains of suckling mice has little relevance for the utilization of the vaccine. To obtain permission for such animal studies the authorities would need a scientific justification for this and why we have selected certain isolates over others. To give an example, in an unrelated project we wanted to test a possible mutation of a genetic marker of a vaccine. So, we are talking about an amino acid exchange of one of the genetic markers of a vaccine candidate due to a mutation of a single nucleotide. After 2 years, we still do not have permission from the authorities to conduct this study. Hence, the probability obtaining permission for the suggested study by the reviewer is almost negligible, although scientifically interesting.    

Reviewer 2 Report

Authors made good effort to address my comments and suggestions.

Author Response

Reviewer agreed to revised version so not point-by-point response necessary